# Becoming Urban Citizens: A Three-Phase Perspective on the Social Integration of Rural–Urban Migrants in China

**DOI:** 10.3390/ijerph19105946

**Published:** 2022-05-13

**Authors:** Xia Sun, Juan Chen, Shenghua Xie

**Affiliations:** 1School of Public Administration, Central China Normal University, Wuhan 430079, China; xia.sun@mails.ccnu.edu.cn; 2Department of Applied Social Sciences, The Hong Kong Polytechnic University, Hong Kong SAR, China; juan.chen@polyu.edu.hk

**Keywords:** social integration, rural–urban migrant, circular migration, urban settlement, urban integration

## Abstract

This article attempts to provide an integrated conceptual framework for understanding how rural–urban migrants in China integrate into urban society. We propose a three-phase conceptual framework in which the social integration of rural–urban migrants is categorized into circular migration, urban settlement, and urban integration. We argue that the three phases differ with respect to the aims of migration, the identity of migrants, the key dimensions of social integration, the role of government, and the *hukou* policy. While the transition from circular migration to urban settlement is an outcome of economic development and change in demographic structure, as reflected by the economic conditions of rural–urban migrants, welfare expansion also plays a critical role during this process. We further hypothesize that the transition from urban settlement to urban integration will be a result of the social interaction between rural–urban migrants and local urbanites, within which social capital and cultural factors are vital. Since most rural–urban migrants are currently at the phase of urban settlement, we suggest that the Chinese government should enlarge welfare provisions to support their settlement in cities. This study contributes to improving the understanding of how to facilitate social integration of internal migrants in developing countries.

## 1. Introduction

International migration from developing countries to developed countries has continued to draw scholarly attention in recent years. With the rapid development of industrialization and urbanization in developing countries, the scale of rural-to-urban migration has also become increasingly prominent. Rural-to-urban migration in China is viewed as the largest population migration in human history [1]. While the social integration of immigrants in developed countries has been widely explored [2,3,4,5], the social integration of domestic migrants in developing countries has received much less attention.

Understanding the process of rural-to-urban migration in developing countries is an urgent task, but one which involves large challenges due to the remarkable social, economic, institutional, and political differences between the countryside and urban areas. In addition, at least three problems have not yet been overcome by the existing studies on rural-to-urban migration in developing countries. First, from a life-course perspective, it is a long process from the time when a rural person decides to migrate to the city to the point at which the person fully integrates into urban society. Thus, it is necessary to carefully consider how this long process should be categorized into different phases in order to simplify understanding. Second, social integration contains various dimensions. Carefully examination should be given to the structure of the social integration of rural–urban migrants, and how this should be understood in a society undergoing rapid social change. Third, given that the social integration of rural–urban migrants is a multistage process and covers various dimensions, it is critical to consider what roles the government should play at different phases of rural-to-urban migration.

In this article, we analyze the case of China and demonstrate the process of rural-to-urban migration from temporary migration to social integration. The paper first briefly reviews studies on the social integration of immigrants in western countries and rural–urban migrants in urban China, indicating that social integration of immigrants is a multistage process, embodies various dimensions, and is influenced by many factors. Based on the review, we establish an integrative conceptual framework and divide the process of rural–urban migration in China into three phases: circular migration, urban settlement, and urban integration. In the next section, we present key determinants for the transition from one phase to another. The theoretical implications of the conceptual framework for future studies are discussed in the final section. Although developed in the context of China’s rural–urban migration, the proposed framework could be generalizable to the urban integration of internal migrants in other social settings, because categorization of the process of rural–urban migration and understanding the role of governments are common issues during rural–urban integration.

## 2. Social Integration of Immigrants in Developed Countries

Social integration can be defined as the process by which immigrants grow to be members of the host society [6] and become capable of participating in a wide variety of social relationships [7]. The social integration of immigrants has been a core social issue and subject of policy debate in industrialized economies such as the European Union, United States, Canada, and Australia, because the number of immigrants in these societies has continued to increase since World War II [8].

The social integration of immigrants was once dominated by the assimilation paradigm, that is, immigrants were expected to abandon their original culture and lifestyle and accept the culture and lifestyle of the host society [9,10,11]. The social integration of immigrants in the United States is a good example of the assimilation paradigm. American society was described as a “melting pot” in which immigrants’ diversities of nationality and ethnicity would vanish during the process of assimilation [12,13,14,15]. Influenced by this theory, the assimilation paradigm was also at the heart of policy implementation in immigrant-receiving societies such as the United States, Australia, and the U.K. during the first half of the 20th century.

However, the assimilation paradigm was challenged both by theory and by policy practice. Segmented assimilation theory, for example, argues that instead of assimilation into the host society, the social integration of immigrants also has other possibilities. This theory distinguishes three different types of social integration outcomes: assimilation, segregation, and marginalization [16,17]. Assimilation means that immigrants integrate into the host society both economically and culturally. Segregation means that immigrants achieve a middle-income class in the host society but deliberately retain their original culture. Marginalization means that immigrants fail to realize upward social mobility while simultaneously losing their cultural identity. In short, they become a marginalized group in the host society.

Multiple possibilities for the social integration of immigrants are also reflected in bi-directional acculturation theory [18,19,20]. This theory argues that immigrants’ acculturation is determined by both the culture of the host society and the culture of the original society. Depending on immigrants’ attitudes toward the two cultures, acculturation outcomes have four possibilities: assimilation, integration, segregation, and marginalization. Assimilation means that immigrants accept the culture of the host society but abandon their original culture; integration means that immigrants accept both cultures; segregation is the opposite of assimilation, meaning that immigrants keep their original culture but reject the culture of the host society; marginalization means that immigrants reject both cultures and, in short, lose their cultural identity in the process of acculturation. Bi-directional acculturation theory further predicts that immigrants who adopt the integration strategy will have the best wellbeing, followed by those who pursue the assimilation and segregation strategies, whereas those who select the marginalization strategy will have the lowest levels of wellbeing [18].

The decline of the assimilation paradigm has also been reflected in policy practice. The rise of multiculturalism in the late 20th century is the best proof of this. A comparative study on the United States, Germany, and the UK shows that with the accumulation of immigrants into these societies, these countries are gradually transforming into multicultural societies [21]. This transformation, then, is mirrored in policies toward immigrants. The Multiculturalism Policy Index shows that multiculturalism policies in many developed countries have persisted and have continued to expand since the 1980s [22]. Moreover, Banting and Kymlicka showed that the majority of developed countries that adopted a multiculturalism policy in the 20th century persisted with this approach in the first ten years of the 21st century [23].

It is noteworthy that social integration embodies various dimensions. The most frequently used indicators include income, employment, culture, identity, civic engagement, and political participation [18,20,24,25,26]. Various measurement scales have also been developed. For instance, the EU indicators of immigrant integration include income, employment, education, health, social inclusion, and civic engagement [5,27]. Harder et al. proposed a multidimensional scale covering six dimensions of integration—psychological, economic, political, social, linguistic, and navigational [4]. When evaluating the social integration of immigrants, it is always beneficial to consider different dimensions. Some immigrants might show particularly positive performance in employment and education, but lesser performance in civic engagement and social participation, while others may be in a completely opposite situation.

Moreover, the social integration of immigrants is not a one-step but rather a multistage process, which means that it is impossible for immigrants to accomplish social integration in a short period of time. Rather, they might need years or even many generations to fully achieve social integration, as has been shown in many studies [2,3,28,29,30], which categorize the migration process into different phases, such as arrival, settlement, and integration. Thus, integration can be viewed as the final step of migration; but before reaching this point, migrants must go through many other steps. Massey describes the scenario of the social integration of Mexican migrants in the United States as follows [28]. First, they migrate from Mexico to the United States, then, as they accumulate experience in the host society, their improved social and economic conditions increase the possibility of their settling down. Over time, family members from abroad arrive to reunite with them. In addition, migrants increasingly build social ties with local people and establish institutional connections. These tendencies lead to a stable and cumulative increase in the likelihood of social integration.

Additionally, the social integration of immigrants can be affected by factors such as demographic traits, socioeconomic conditions, social relations, the natives’ attitudes toward immigrants, and migration policies. For instance, the political integration of immigrants is influenced by their social capital [24,31,32]. The attributes of immigrants, including language, religious belief, and educational attainment, along with government policies toward immigration, significantly affect the social identity of immigrants [25,26,33]. Furthermore, some studies have suggested that individual-level factors and external factors often interact with each other, mutually determining the social integration of immigrants [25,33]. Since the social integration of migrants is a multidimensional process and is influenced by many factors, Mabogunje proposes a systems approach which considers the migration process to be a nonlinear, symbiotic, increasingly complex, and self-modifying system [34].

## 3. Social Integration of Rural–Urban Migrants in China

Rural–urban migrants are a large and special group in urban China. The magnitude of this group was reported to have reached 286 million in 2020, among whom approximately 170 million were cross-town migrants [35]. This group is unique, because the household registration (*hukou*) system divides Chinese citizens into two categories: agricultural (rural) and nonagricultural (urban).

*Hukou* is an official document issued by the Chinese government. The function of the *hukou* is to certify that the holder of the document is a legal resident of a place. Established in 1958, the *hukou* system is a core institution that defines the relationship between urban and rural areas. Under this system, there are huge differences in the political and economic rights of urban and rural populations [36]. People with an urban *hukou* usually enjoy better social welfare rights, such as medical care, public housing, and children’s education, than people with a rural *hukou*. Furthermore, the conversion of *hukou* status from rural to urban is very difficult, especially concerning the attainment of a local urban *hukou* in megacities such as Beijing, Shanghai, and Shenzhen. Since rural–urban migrants only have non-local rural *hukou*, they are not treated equally as urban citizens by local governments. In terms of their livelihoods and wellbeing they are poorer than the local urbanites [37,38].

The unique social and institutional environment in urban China means that the social integration of rural–urban migrants is a special case, compared to international migrants in developed countries. Usually, immigrants in developed countries are different from locals in terms of skin color, religion, ethnicity, or national identity. However, rural–urban migrants in China share many things in common with urban locals in these aspects [39]. On the other hand, the unique *hukou* system means that rural–urban migrants must break the institutional barrier to integrate into the city. Thus, the *hukou* system can serve as a lens through which to look at the social integration of migrants in the Chinese social context.

In China in recent years, methods have been widely discussed and debated for promoting the social integration of rural–urban migrants in cities. Indeed, this topic is important, because this great migration has been the most significant ongoing demographic procedure in China since the 1980s, and has profoundly affected many aspects of Chinese society [38,40]. Meanwhile, the monitoring and control of rural-to-urban migration have been major national policy concerns in China. Since the 1980s, the central government has implemented extensive policy documents to guide the process both directly and indirectly. All these policies serve a common goal: to achieve controllable rural-to-urban migration.

While the Chinese government is very concerned about the social integration of rural–urban migrants, numerous researchers from multiple disciplines have paid it tremendous attention. Scholars have evaluated the social integration of rural–urban migrants on the basis of different dimensions, including economic integration [41,42,43], urban settlement [44,45,46,47,48,49,50,51,52,53,54,55,56,57], neighborhood relations [39,58], spatial integration [59,60,61,62], acculturation [63], and identity [42,64,65]. Great efforts have been made to examine the factors influencing the social integration of rural–urban migrants [38,41,42,43,66,67,68]. A common finding of these studies can be summarized as follows: although increasing numbers of rural–urban migrants intend to permanently settle in cities, their social integration is at a relatively low level, and they are still confronted with institutional barriers, cultural obstacles, social welfare discrimination, and labor market segmentation, impacting their full integration into urban society. Such findings indicate that China still has a long way to go before it achieves true population urbanization.

Although many studies have reported the difficulties faced by rural–urban migrants in integrating into Chinese urban society, there are also studies which suggest that rural–urban migrants have been gradually transitioning into the urban middle class [41]. The differences in conclusions may be due to the distinctive dimensions of social integration that were examined. In addition, differences in research subjects can lead to different findings. For instance, Tang and Feng reported that the social integration of second-generation rural–urban migrants has been better than that of first-generation rural–urban migrants [56].

The difficulty for the social integration of rural–urban migrants is deeply related to social exclusion in Chinese cities. Chow and Lou summarized four types of exclusion for rural–urban migrants in cities: institutional exclusion, labor market exclusion, exclusion in social relationships, and discrimination and stigmatization [68]. Because of the extensive social exclusion in urban China, rural–urban migrants have a long way to go until they can fully integrate into cities.

The *hukou* system is the most widely recognized form of institutional exclusion. This system has created a dual society that divides China into “one country, two societies” [69]. The function of *hukou* has been widely documented, and includes categorizing Chinese people into rural and urban, boosting urban industry, controlling population mobility, maintaining social and political stability, and determining welfare allocation [37,69,70,71,72,73]. Because of this system, rural people, including rural–urban migrants, are treated as a “second-class population” compared to urban residents [69,74]. The *hukou* system seriously affects the mental health of rural–urban migrants [75,76]. As a result, it is extremely difficult for rural–urban migrants to break the institutional barriers and integrate into urban society.

Employment exclusion limits opportunities for rural–urban migrants to obtain good jobs in the labor market. They are often limited to jobs with dangerous, dirty, and demeaning aspects [37,77]. Consequently, rural–urban migrants are paid less than urban residents [1,78,79,80,81,82], and at the same time, their health status can easily deteriorate due to their high-risk work, which may force them to return to their hometown. This could exacerbate the contradiction between the allocation of medical resources and demand in rural and urban China, further intensifying the already widening health status gap between rural and urban residents [83]. Undoubtedly, the limited job opportunities and low income restrict the economic integration of rural–urban migrants. Additionally, low socioeconomic status limits rural–urban migrants’ social interaction with urban locals, which creates a barrier to other aspects of social integration, such as community participation, civic engagement, and urban identity.

Exclusion from social relationships is reflected in the lack of social interaction between rural–urban migrants and local urbanites [65]. A notable reason for exclusion from social relationships is due to residential segregation. Rural–urban migrants tend to reside in urban villages located in the marginal areas of cities [62]. The urbanized environment reflects a lack of synchronization between migrants and their resettlement community environments [84]. Although these groups often show strong intentions towards intragroup and intergroup social interactions [39,58,85], urban governments’ extensive redevelopment schemes tend to impede their social integration by limiting both intragroup and intergroup social ties [58,86].

Despite their great contribution to economic development, rural–urban migrants have suffered serious social discrimination and stigmatization in urban areas [79,87,88,89,90]. This is why they are labeled a “second-class population”. In addition, discrimination and stigmatization are negatively associated with the health and subjective well-being of rural–urban migrants [91,92,93,94], which is detrimental to their psychological integration in cities.

Other studies have examined factors affect the social integration of rural–urban migrants. The main factors tested include socioeconomic characteristics, social capital, and cultural capital [42,43,65,66]. These studies have usually reported positive associations. Nonetheless, there is often a gradient of effects regarding the influence of different factors on the social integration of rural–urban migrants. For instance, Wang and Ning found that factors relating to destination exert a stronger impact on the social integration of rural–urban migrants than individual-level factors [43]. Yue et al., report that the effect of social ties with local urbanites is stronger than social ties with other migrants in promoting the social integration of rural–urban migrants. Noticeably, even though socioeconomic factors, social capital, and cultural capital are positively related to the social integration of rural–urban migrants, the effects of these factors are limited due to the pervasive existence of social exclusion in Chinese cities. Rural–urban migrants are still in danger of falling into the urban underclass [65].

Prior studies have contributed significantly to our understanding of the social integration of rural–urban migrants. However, to date, there is still no integrated framework for conceptualizing the social integration of rural–urban migrants in China. Moreover, previous studies have often adopted a static perspective from which to understand the social integration of rural–urban migrants, making difficult the comparison of results across studies. An integrated conceptual framework should thus employ a dynamic perspective and cover certain significant dimensions that have been ignored in previous studies.

## 4. Social Integration of Rural—Urban Migrants: A Multistage and Dynamic Process

The experience of international migrants indicates that social integration is a multistage process. Therefore, it is of great importance to use a multistage and dynamic perspective to understand the social integration of rural–urban migration in China. Previous studies have focused on the social integration of rural–urban migrants to determine what factors affect their social integration. However, from a multistage and dynamic perspective, comparisons of research findings between different studies is extremely difficult. For instance, while some studies have found that the neighborhood cohesion of rural–urban migrants is better than that of urban locals [95], others found that they generally feel socially excluded [61]. The former group of studies may reach the conclusion that the social integration of rural–urban migrants is good, but the latter group of studies may reach the opposite conclusion. In fact, a premise of the comparison is to adopt a multistage and dynamic perspective and to know to which stage of social integration the rural–urban migrants belong, and then to decide which indicators should be used for comparison.

Additionally, previous studies have neglected to consider changes during different periods of social integration in rural–urban migrants’ intentions and identities, dimensions of integration, the role of the government, and the *hukou* system. An integrated conceptual framework must incorporate these dimensions and employ a dynamic perspective to explain how these aspects change throughout the process of social integration.

### 4.1. Aims of Rural–Urban Migrants

The intentions of rural–urban migrants can change during different phases. Economists often view migration as a rational behavior that migrants undertake in order to maximize their income [96,97,98,99,100]. This might be true in the early stages of rural–urban migration. However, there are things other than money for migrants to pursue once they have achieved decent incomes. Home ownership, citizenship, family union in the host society, and social participation can over time also become the aims of migration. It is important to identify what rural–urban migrants’ main intentions will be at different stages of rural–urban migration.

### 4.2. Identity of Rural–Urban Migrants

Identity can be defined as how people categorize themselves as belonging to a specific social group [101,102,103,104]. The identity of rural–urban migrants is changeable throughout the process of urban integration. Theoretically, the longer that rural–urban migrants stay in cities, the more likely they are to identify themselves as urban citizens, because they have had more time to obtain good jobs, improve their housing conditions, establish social relationships with locals, and participate in local activities. Therefore, the identity of rural–urban migrants changes from being urban sojourners to becoming urban citizens.

### 4.3. Dimensions of Social Integration

The social integration of migrants encompasses a wide variety of dimensions, such as economic integration, occupational integration, spatial integration, acculturation, civic engagement, and identity. However, studies on the social integration of rural–urban migrants in urban China have often excessively simplified the measurement of rural–urban migrants’ social integration. Most of these studies have focused on socioeconomic integration, acculturation, and identity [42,43,65,66]. Other dimensions, such as civic engagement, political integration, and community participation, have received scant attention. This imbalance in attention allocation is not conducive to a systematic understanding of the social integration of rural–urban migrants. Therefore, this article attempts to establish an integrated conceptual framework to include different dimensions such as socioeconomic integration, acculturation, identity, civic engagement, political integration, and community participation.

Another significant issue concerning the measurement of the social integration of rural–urban migrants is that previous studies have normally regarded different dimensions of integration as parallels, while neglecting the truth that social integration can be progressive and transitional. In other words, migrants might have better outcomes in some dimensions and worse in others, often depending on how long they have been in the city. The most recent empirical study, based on the CGSS 2012–2013, reveals that structural assimilation—the socioeconomic integration of rural–urban migrants—takes place before extrinsic acculturation, such as speaking the local language [65]. Studies, therefore, suggest that it is not correct to treat different dimensions of the social integration of rural–urban migrants as parallels. Rather, there may be an order of occurrence in different dimensions of social integration. Moreover, to regard the social integration of migrants as multiple processes is not a new approach. Gordon categorized assimilation into seven procedures: acculturation, structural assimilation, marital assimilation, identification assimilation, attitude reception assimilation, behavior reception assimilation, and civic assimilation [9]. Similarly, Park divided the assimilation of immigrants in the United States into four steps: contact, competition, accommodation, and assimilation [105]. It is reasonable to assume that the social integration of rural–urban migrants in China also follows a multidimensional process.

### 4.4. The Role of the Government

What roles should government play in the social integration of rural–urban migrants? Previous studies have provided valuable policy recommendations such as improving housing conditions [45,54,106,107], rethinking urban redevelopment schemes [58,59,108], and expanding welfare provisions [109,110]. These recommendations are indeed very important. However, when should a specific policy be implemented? Given that the social integration of rural–urban migrants is not a static process, and different dimensions of integration are not parallel, the role of government should change at different phases. Previous studies on the topic have often neglected this point.

### 4.5. The Hukou System

How should government reform the *hukou* system? Undoubtedly, the *hukou* system should be a policy focus of the government, because it is a key source of social exclusion and discrimination in Chinese cities. However, *hukou* reform is a complex project and cannot be accomplished within a short period. The best choice for the government is to adopt a strategy of gradual reform. An integrated conceptual framework should be able to incorporate gradually reform of the *hukou* policy at different stages of the social integration of rural–urban migrants.

## 5. Three Phases of Social Integration of Rural–Urban Migrants in China

We propose a three-phase conceptual framework for understanding the social integration of rural–urban migrants in China. Figure 1 shows the basic structure of the conceptual framework. First, a multistage and dynamic perspective divides the social integration of rural–urban migrants into three phases: circular migration, urban settlement, and urban integration. Second, the conceptual framework covers the aims of rural–urban migrants, the identity of rural–urban migrants, the dimensions of social integration, the role of government, and changes to the *hukou* system, and uses a dynamic perspective to analyze how these dimensions change in different phases. The conceptual framework hypothesizes that rural–urban migrants tend to show common patterns during different periods. This is very important, because if policy recommendations are to be made at the macro-level to improve the social integration of rural–urban migrants, it is necessary to capture the social integration pattern of the majority of rural–urban migrants. Thus, this framework proposes that circular migration was the most common pattern from the 1980s until the 2000s, urban settlement will be the most common pattern from the 2000s until the 2030s, and urban integration will be the most common pattern after the 2030s. We present the traits of each phase in successions.

### 5.1. Circular Migration

The phase of circular migration is characterized by rural–urban migrants traveling between their hometown and the host city. The most important reason for rural–urban migrants migrating to the city is to earn money to support their families in the countryside. Thus, income and employment are the two most important indicators for measuring the social integration of rural–urban migrants. During this phase, rural–urban migrants are regarded by urban governments as urban sojourners, and the countryside is the place to which they will finally return.

During the phase of circular migration, the policy focus is control. The Chinese government gradually relaxed the very strict *hukou* policy so that rural people have been able to move to cities to find jobs. However, during this phase, the control of population mobility was still very strict. In particular, the *hukou* system remained closely associated with the social rights of Chinese people. Therefore, although rural–urban migrants came to live in cities, urban governments treated them as outsiders in the city and granted them few welfare entitlements.

Circular migration was the most significant pattern of rural–urban migration in China from the 1980s until the late 2000s. Table 1 summarizes the choices that rural–urban migrants intended to make between circular migration and urban settlement according to surveys conducted by different research projects in China [44,45,46,47,48,50,51,52,56,111,112,113,114]. If we use 50 percent as the threshold to judge the main pattern of rural–urban migration in China, 2008 can be regarded as the watershed moment. Before 2008, less than 50 percent of rural–urban migrants intended to permanently settle in cities; the majority wished to straddle the countryside and the city. However, after 2008, the proportion of rural–urban migrants who intended to permanently reside in cities was consistently above 50 percent, with the exception of the study by Yang and Guo [48].

### 5.2. Urban Settlement

Skeldon suggested that circular migration as a migration pattern will give way to urban settlement when the urbanization ratio reaches a certain point [115]. The phase of urban settlement is characterized by the decision of rural–urban migrants to settle in the city. During this phase, with the accumulation of experience in cities, there has been a gradual increase in the income of rural–urban migrants. Because of this, rural–urban migrants have become more inclined to bring family members from the countryside to live in the cities. To encourage rural–urban migrants to stay permanently in cities, it is very important for them to be granted social welfare entitlements and to be able to obtain decent housing. Thus, welfare rights and housing conditions can be viewed as vital indicators of social integration during the phase of urban settlement. Additionally, during this phase, rural–urban migrants were unlikely to think of themselves as urban sojourners. Rather, they increasingly treat themselves as permanent residents in cities.

During the phase of urban settlement, the policy focus has been on welfare provision. During this phase, the Chinese government started to reform the *hukou* system, aiming to reduce its restrictions on the urban settlement of rural–urban migrants. However, rural–urban migrants also need social welfare to protect them against risks in cities. During the phase of urban settlement, the distribution of rural–urban migrants in cities of different scales has been very unequal, with too many in the large cities and too few in small cities and towns. Radical elimination of the *hukou* system is impossible during the phase of urban settlement, especially in megacities. Song suggests that in mega-cities, the attractiveness of the city caused by wage premiums is not able to offset the combined repellant force caused by high housing prices, bad urban social networks, air pollution, and health deterioration [116]. Thus, welfare provision has become a pressing issue. The Chinese government must remove the current association between welfare rights and local *hukou* registration. Even if rural–urban migrants have no local *hukou*, they should also enjoy basic welfare rights in the host city.

During the phase of urban settlement, the Chinese government has adopted the strategy of category management, as reflected by the *National New-Type Urbanization Plan 2014–2020*, which states that the basic principle of *hukou* reform is “completely eliminating restrictions on rural people seeking to settle in small cities and towns, eradicating barriers on settling in medium-sized cities, rationally determining the criteria for settling in large cities, and strictly controlling the population of megacities”. The policy indicates that the Chinese government hopes to utilize the *hukou* policy to adjust the distribution of the population among cities of various scales in order to achieve an equal distribution of the population in China.

Urban settlement will be the most significant characteristic of rural–urban migration in China between the late 2000s and the 2030s. Our judgment is based on three reasons. First, the change in rural–urban migrants’ salary potential explains why the phase of urban settlement arrived in the late 2000s, with China meeting the so-called Lewis turning point in this period. According to Lewis, during the early phase of industrialization and urbanization of a country, because there is unlimited labor supply, rural labor wages will be very low [117]. Then, with continued development, the rural labor pool will gradually become exhausted, bringing about a shortage of labor and the growth of wages. This situation is exactly what happened in China [118,119,120,121]. Figure 2 shows the wage trend of rural–urban migrants between 1979 and 2016 [122,123]. A relatively slow increase can be seen before 2008, while a rapid increase can be seen after that time point. The increase in the wages of rural–urban migrants indicates that the rural labor pool in China has been gradually exhausted, and it is possible for the migrants to settle in the cities.

Second, the Chinese government has an ambition to promote the urban settlement of rural–urban migrants. For instance, the *National New-Type Urbanization Plan 2014–2020* stated that the Chinese government will promote the settlement of at least 100 million from the rural population into cities by 2020. Chan describes this as a new blueprint of urbanization in China. If this trend continues after 2020, the migrant pool in cities will be substantially exhausted [124].

Third, the World Bank and the Development Research Center of the State Council launched a report, *China 2030: Building a Modern, Harmonious, and Creative Society*, which predicts that the Chinese government will offer residence permits to all residents in cities by 2030, to enable rural–urban migrants to enjoy equal social rights to those of local urbanites and to permanently reside in cities [125]. Therefore, we maintain in our conceptual framework that the phase of urban settlement extends from the late 2000s to approximately the 2030s.

### 5.3. Urban Integration

The phase of urban integration is followed by the phase of urban settlement. By then, even though rural–urban migrants have achieved settlement in cities, they are far from being fully integrated into the city. First, although an increase in income enables them to permanently live in cities, the income gap between rural–urban migrants and local urbanites still exists. Rural–urban migrants are still limited to working in certain labor sectors. The share of employees with a background of rural–urban migration in labor sectors of high occupational status is still lower than that of those who were born in cities. Second, rural–urban migrants may continue to face unequal opportunities in many areas, such as children’s education, employment, and healthcare services, compared to local urbanites. Equal opportunity is a very important indicator for evaluating the social integration of immigrants [12,104,126]. Third, although the rural–urban migrants have settled in cities, they may be clustered in migrant enclaves in their host city. They need to spatially assimilate into the host city in order to establish more diversified social networks. Fourth, the voice of rural–urban migrants in social and political fields remains weak. The rate of civic engagement, political participation, and community participation is lower than for local urbanites. Entering the phase of urban integration, rural–urban migrants still have much to achieve in order become real urban citizens. Therefore, indicators such as spatial integration, acculturation, urban identity, and political integration are important dimensions of rural–urban migrants’ social integration.

During the phase of urban integration, the role of the government is to promote complete social integration of rural–urban migrants. The Chinese government should adopt a “cooperative governance” strategy during this phase. This strategy means that the government acts together with the market, NGOs, and volunteer individuals to promote the social integration of rural–urban migrants. Among these actors, rural–urban migrant participation is very important, because it will help them to learn the local language, build social ties with local urbanites, and, more importantly, reinforce their sense of urban belonging. The joint action of different agencies may also improve social solidarity and help to rebuild civil society in China.

During the phase of urban integration, the “elimination” of the *hukou* system becomes possible. Notably, “elimination” does not necessarily mean that there will be no *hukou* system anymore. Instead, it means that *hukou* will go back to its real function, namely, population registration. Many other functions associated with the *hukou*, such as population categorization, welfare allocation, and population mobility control, will be abolished. This step is very important to achieving the goal of equal citizenship in China.

We predict that urban integration will become the main issue of rural–urban migration after the 2030s. By that time, the majority of rural–urban migrants will have settled in cities and become permanent urban citizens. However, settling in cities does not guarantee that rural–urban migrants will have fully integrated into urban society. At present, their political participation, civic engagement, and urban identity are far worse than those of the urban locals. Furthermore, their poor social participation is not a phenomenon unique to domestic migrants in China, but is also the case in other social settings. Zimmer found that in the United States, compared to the native population, migrants as a group have a much lower rate of social participation measured by participation in formal organizations, officership, and registration to vote, but that this gap becomes smaller between second-generation migrants and the native population [127].

## 6. Determinants of Phase Transition

The conceptual framework proposed in this study divides rural-to-urban migration into three phases. However, what are the determinants of transition from one phase to another? There are many classical theories to explain the determinants of rural-to-urban migration, such as the dual economy theory, the pull–push theory, the new economics of migration theory, and the Todaro–Harris model [128,129,130,131,132,133,134,135]. These theories emphasize factors which determine rural migration to urban areas, which has also been carefully examined in the Chinese social context. However, as suggested in this study, rural-to-urban migration spans more than two phases, and these theories do not give enough attention to what determines the transition from one phase to another. This question is very important if we assume that governmental strategies toward migrants should be distinctive at different phases.

First, the transition from circular migration to permanent urban settlement is an outcome of economic development and change of demographic structure, as reflected by the economic conditions of rural–urban migrants; meanwhile, welfare expansion also plays a critical role during this process. Decent income is a premise for migrants to permanently stay in cities. However, the income of rural–urban migrants, on the one hand, relies on economic development; on the other hand, this income also hinges on the change of demographic structure. The Lewis model suggests that even in a rapidly developing economy, if the labor pool in the countryside is unlimited, the wages of rural–urban migrants will not increase. Only when the labor pool in the countryside is near exhaustion will the wages of rural–urban migrants substantially increase.

In addition to the importance of economic conditions, welfare expansion is also very important for the urban settlement of rural–urban migrants. Migration is highly selective [136], and this is the case in China. Zhang and Wang suggested that the possibility for rural–urban migrants to obtain the urban *hukou* is unlikely to be based on the length of their urban residence, but more likely on their contribution to the host city [137]. Thus, unless the government tries to establish a universal social welfare system that also covers rural–urban migrants in cities, it will be extremely difficult for them to permanently live there.

During the period of transition from circular migration to urban settlement, welfare coverage in China has greatly expanded. Welfare expansion in China is reflected by Article 3 of the Social Insurance Law of China, which states that the law follows the principle of “wide coverage, modest benefits, multi-tiered programmes”. This means that the Chinese government has set the goal of gradually including rural–urban migrants into the urban social welfare system, thus enabling them to permanently settle in cities. In addition, the reform of the *hukou* system in recent years has aimed to remove the bond between welfare rights and the *hukou*. Yao, Liu and Sun note that inequalities remain in both medical and public health services between internal migrants with and without local health insurance coverage, and that it is necessary to coordinate the relationship between the *hukou* system and welfare rights [138]. In short, even if rural–urban migrants have no local *hukou*, they may still be able to enjoy the social welfare entitlements of the host city.

Second, the transition from urban settlement to urban integration is an outcome of social interaction between migrants and urban residents, within which social capital and cultural factors are vital. Social capital refers to “features of social organization such as networks, norms and social trust that facilitate coordination and cooperation for mutual benefit” [139]. Social capital plays a significant and positive role in many social areas, such as education, employment, crime control, life satisfaction, and poverty reduction [140,141,142,143,144,145]. According to Putnam, social capital is beneficial because networks of civic engagement established through social capital form reliable rules of generalized reciprocity and thus bring about the formation of social trust [139]. Similarly, social capital is also beneficial for rural–urban migrant integration into urban society, as it may facilitate a stronger connection with local urbanites, and thus encourage social trust between the two groups.

The social integration of migrants is closely related to social capital [65]. However, rural–urban migrants first need to settle in cities and gradually formalize steady social networks to establish connections with local urbanites before they can exhibit more social participation. Some studies have found that social capital is not significantly associated with rural–urban migrants’ permanent urban settlement intention [47], but with their urban integration [65]. Moreover, Wang and Fan found that income was only partially associated with the social integration of rural–urban migrants [42]. Other conditions being equal, rural–urban migrants may either integrate into cities or reconstruct their rural lifestyles there, and social and cultural factors are crucial to that choice [146]. These studies indicate that social capital, rather than socioeconomic status, is more important during the phase of urban integration of rural–urban migrants.

The role of cultural factors in the social integration of rural–urban migrants should also be emphasized. Chinese cities have large variations in dialects, culture, and lifestyles, which certainly affect the social adaptation and integration of rural–urban migrants. Wang and Fan found that rural–urban migrants who spoke a local language and were socially and culturally adapted tended to have better integration outcomes [42]. After urban settlement, rural–urban migrants, and their offspring in particular, will gradually learn the local culture and lifestyles, which acts as a bridge to their participation in local activities. Thus, cultural adaptation should be regarded as an essential element of social integration.

According to the conceptual framework, most rural–urban migrants in the 2020s are currently in the phase of urban settlement. Thus, the Chinese government should increase welfare provisions to rural–urban migrants to enable their settlement in cities. Present government policy still has its weaknesses. For instance, Chan notes that the implementation of the *National New-Type Urbanization Plan 2014–2020* gives priority to those migrants who are well-educated, highly skilled, and wealthy, while others may find it harder to obtain the urban *hukou*, along with its associated welfare entitlements [70]. Furthermore, Zhang and Wang suggest that urban governments are highly selective, and only preferred rural–urban migrants are likely to be selected for the urban social welfare system [137]. Therefore, it is essential for the Chinese government to adjust its policies to keep pace with the course of rural–urban migration in China.

## 7. Discussion and Conclusions

This article reviews extant studies on the social integration of immigrants in developed countries and rural–urban migrants in China and proposes an integrated conceptual framework that divides the social integration of rural–urban migrants into three phases. This is in line with the study by Khalid and Urbański, which revealed that migration is a multistage process and that governments can ease the migration process through support policies and regulations [147]. Our study further examines the key determinants of phase transition and addresses policy foci during the current phase of rural–urban migration. Policy recommendations for improving the social integration of rural–urban migrants are proposed at the macro level, and it will be necessary to capture the social integration pattern of most rural–urban migrants. The paper does not consider the fact that these three phrases may be interrupted or regress during the migration process. Nonetheless, the study should have significant implications for future studies of rural–urban migration in China.

First, social integration is a multistage and dynamic process, which means that when examining the social integration of rural–urban migrants, researchers should determine to which phases of rural–urban migration the investigated migrants belong. Determination of these phases is very important in reaching the conclusion of a study. For instance, if migrants are at the phase of circular migration, they usually show a low rate of political participation. Therefore, even if such a study finds that the political integration of rural–urban migrants during this phase is not high, the conclusion should not necessarily be pessimistic. Rather, researchers should view this as a necessary step of integration during which migrants will experience a low rate of political participation before closer assimilation into the host society.

Second, future studies should pay more attention to social integration as a multidimensional concept that includes aspects such as income, employment, welfare rights, housing conditions, civic engagement, and political participation. Previous studies have usually focused on the socioeconomic integration of rural–urban migrants. To obtain a systemic view of the social integration of rural–urban migrants, other integration dimensions such as civic engagement and political participation should also be carefully considered and examined.

Third, since the key dimensions of social integration during different phases of rural–urban migration vary, an integrated social integration scale with phase characteristics is proposed. A previous study proposed a multidimensional measure of migrant integration [4]. However, this measure treated all dimensions as being equal and existing in parallel, which may not be suitable for the Chinese context. An integrated social integration scale with phase characteristics could draw on the Multidimensional Poverty Index, which aims to measure the extent to which people suffer multiple deprivations as well as the intensity of such deprivations [148]. Specifically, when evaluating the social integration of rural–urban migrants during the early phase of migration, more weight could be assigned to employment, income, and housing conditions, while during the later phase of migration, more weight could be assigned to civic engagement, identity, and political participation. The integrated multidimensional scale should be able not only to include different dimensions of social integration, but also to consider the relative importance of different dimensions according to the phase of rural–urban migration.

Fourth, future studies should also pay attention to the changing role of the government, as well as *hukou* system reform, in the process of the social integration of rural–urban migrants. During the phase of circular migration, the government tended to control the order of rural–urban migration. Therefore, the *hukou* policy was only gradually relaxed to allow population mobility. When the migration pattern starts to transition from circular migration to permanent urban settlement, the main responsibility of the government is to secure the livelihoods and wellbeing of rural–urban migrants in cities. Therefore, welfare provision will become key in government policy. In particular, the government will start to reform the *hukou* system in order to increase social welfare entitlements for rural–urban migrants. This is similar to the situation of migrants in developed countries. For instance, Máté, Sarıhasan and Dajnoki found that labor market institutions such as the minimum wage, unemployment benefits, union density, and active labor market policies can significantly affect benefits for both native and international migrants [149]. During the phase of urban integration, those who have settled in cities pursue participation in civic and political activities. Therefore, the Chinese government should adopt a “cooperative governance” strategy during this phase. Furthermore, during this phase, the “elimination” of the *hukou* system becomes possible. *Hukou* may go back to its real function of population registration, which exists in many countries. It should be noted that although this study has emphasized that the role of the government should change according to the dynamic process of migration, this does not mean that the government should pay attention only to the demands of the majority of rural–urban migrants during each phase. Rural–urban migrants lagging behind the macro process of migration should also be a policy focus of the government.

Finally, since most rural–urban migrants are currently in the phase of urban settlement, it is very important for the Chinese government to improve the urban social welfare system so that rural–urban migrants will be able to permanently settle in cities. Therefore, future studies should explore how to improve the current Chinese social welfare system and enhance the welfare conditions of rural–urban migrants.

## Figures and Tables

**Figure 1 ijerph-19-05946-f001:**
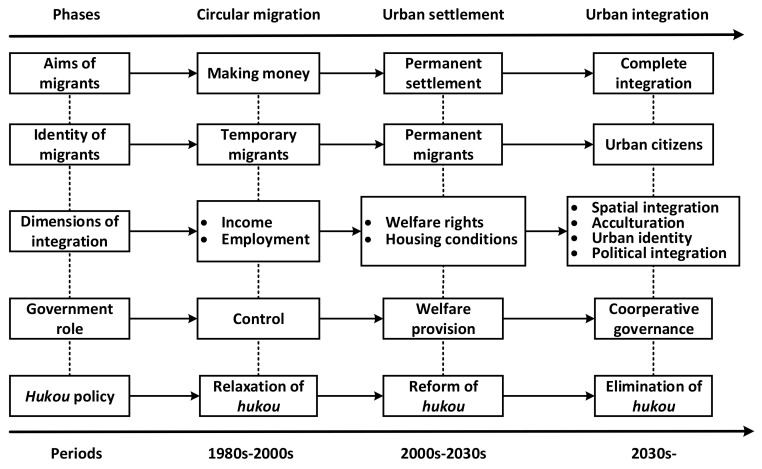
The conceptual framework for understanding social integration of rural–urban migrants.

**Figure 2 ijerph-19-05946-f002:**
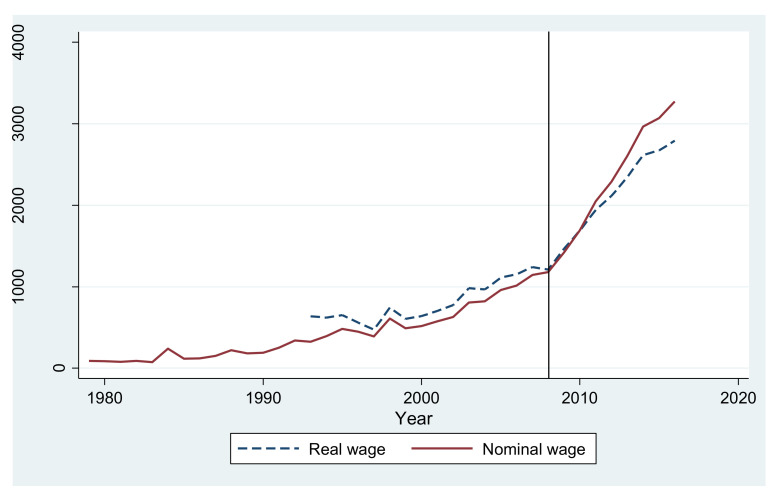
The wage trend of rural–urban migrants in China, 1979–2016. Source: (Lu 2012) and (National Bureau of Statistics of China 2017) (The real wage is calculated on the basis of the consumer price index from OECD (2014 = 100)).

**Table 1 ijerph-19-05946-t001:** Urban settlement intentions of rural–urban migrants according to different surveys.

Authors	Survey Regions	Survey Year	Settlement (%)	Source
Zhu	5 cities in Fujian Province	2002	21%	[111]
Zhu and Chen	6 cities in Fujian Province	2006	36%	[51]
Fan	50 urban villages in Beijing	2008	38%	[44]
Chen and LiuLiu and Wang	12 cities in the Yangtze Delta, the Pearl River Delta, the Bo-Hai Rim and the Chengdu-Chongqing region	2009	55%	[47,112]
Cao et al.	12 cities across four major urbanized regions of China	2009	52%	[46]
Hao and TangTang and Feng	13 prefecture-level cities, 52 county-level cities and 27 townships in Jiangsu Province	2010	52%	[50,56]
Tan et al.	31 provinces, autonomous regions and municipalities	2012	60%	[113]
Xie et al.	15 cities in the eastern, central and western areas of China	2008–2009	61%	[52]
Yang and Guo	Ningbo city, Zhejiang Province	2014	48%	[48]
Huang et al.	Eight cities in China	2013	55%	[114]

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
