# Peer review of "Becoming Urban Citizens: A Three-Phase Perspective on the Social Integration of Rural–Urban Migrants in China"

_ijerph, 2022, doi:10.3390/ijerph19105946_

Round 1
Reviewer 1 Report
The article, a theoretical reflection on rural-urban Chinese internal migration, is conceptually well organized and linguistically clear. The theoretical frameworks are adequate. For these reasons it can be a useful contribution to the national and international debate on how to facilitate social integration in migration.
From the theoretical point of view, I consider 2 aspects to be discussed. First of all, the idea that migration process has a linear development that takes place in progressive steps: from circularity, to settlement and to social integration. These phases are not often linear and gradual but, depending on specific situations, there may be interruptions and/or regressions in the migration process. However, the conceptual aspect most questionable is the idea of ​​social support for integration according to the migratory phase. This hypothesis risks to slow down and stop the dynamic process of migration, offering fewer social guarantees to those who are already in a precarious condition in terms of income, education, bonds and social interactions.
Anyway, in my opinion the article deserves to be published in order to improve the debate on social integration in migration process.
Author Response
We sincerely appreciate all these helpful comments kindly offered by the reviewer, which help broaden our analytical horizon and further sharpen the contribution of this work. Based on these comments, we focused on two major aspects when revising the paper.
Firstly, we totally agree that there may be interruptions and/or regressions in the migration process. But considering the need to make policy recommendations at the macro-level to improve the social integration of rural-urban migrants, it is necessary to capture the social integration pattern of most rural-urban migrants. Although this paper has not considered the fact that these three phrases may be interrupted or regressed in the migration process, we believe that the final conclusions drawn in this paper for the three phases still apply.
Secondly, we added in the revised manuscript: It should be noted that although this study emphasized the role of the government should change according to the dynamic process of migration, it does not mean that the government should only pay attention to the demands of the majority of rural-urban migrants at each phase. Rural-urban migrants lagged behind the macro process of migration should also be the policy foci of governments. (please see page 15, line 655-661)
Reviewer 2 Report
The authors should reconsider rephrasing the first sentence of the manuscript. The sentence seems to be too long. Breaking it down into more than one long sentence will help in improving readability and better understanding.
For in-text citations shown as [2,3,4,5] they can be represented as [2-5] as recommended by the journal, also in lines 70, 73, 81, 88, 113[…24-26], 125[…28-30], 180. Please this occurs throughout the text, effect the corrections.
In line 67, consider changing ‘rapidly’ to ‘after’ to show that the trend of migration increased after WWII and is still growing.
Expatiate on the hukou system for readers to know what you are talking about or its implications. The hukou system first appears in line 165, no definition or explanation of the system was given.
Kindly define and explain before talking about its impact. The authors need a section devoted to explaining the hukou system.
The declaration in line 200 about the difficulty in the social integration of migrants is not rooted in the literature.
Please add some empirical discussion that supports your view.
Line 268 – Did you mean to write ‘socially’ excluded? Make the corrections.
Line 281 – Rephrase to maximize their income.
After the observations you made from line 303, what does your study hope to add to correct the imbalance in studies with regard to your observations?
This should come next before you start making further revelations of shortcomings in the study of urban-rural migration.
The sentence from 304-to 308 should be broken up into two or more sentences to improve their readability.
Line 308 about the study by CGSS in 2012-2013 is not recent; rather authors may say that ‘the most recent CGSS study in 2012-2013.’ Recast that line.
Please also consider to cite the following article in the Discussion and Conclusion
Khalid, B., & UrbaÅ„ski, M. (2021). Approaches to understanding migration: A multi-country analysis of the push and pull migration trend. Economics & Sociology, 14(4), 242–267. https://doi.org/10.14254/2071-789x.2021/14-4/14
Máté, D., Sarihasan, I., & Dajnoki, K. (2017). The relations between labour market institutions and employment of migrants. Amfiteatru Economic, 19(46), 806–820.
The entire study requires extensive editing and proofreading to improve the grammar and flow of the manuscript. Thank you.
Author Response
1 The authors should reconsider rephrasing the first sentence of the manuscript. The sentence seems to be too long. Breaking it down into more than one long sentence will help in improving readability and better understanding.
Response: Thank you very much to point out the sentence structure issue in our manuscript. According to this comment, we have shortened the first sentence of the manuscript. (please see page 1, line 26-30)
2 For in-text citations shown as [2,3,4,5] they can be represented as [2-5] as recommended by the journal, also in lines 70, 73, 81, 88, 113[…24-26], 125[…28-30], 180. Please this occurs throughout the text, effect the corrections.
Response: Thank you for this comment. We have double-checked and revised the citations according to the recommendation.
3 In line 67, consider changing ‘rapidly’ to ‘after’ to show that the trend of migration increased after WWII and is still growing.
Response: Thanks for your comment. According to this comment, we have revised the sentence to make it clearer. (please see page 2, line 67-68)
4 Expatiate on the hukou system for readers to know what you are talking about or its implications. The hukou system first appears in line 165, no definition or explanation of the system was given.
Response: Thanks for this comment. We have provided a definition of hukou in the footnote according to your comment. (please see page 4, line 155-159)
5 Kindly define and explain before talking about its impact. The authors need a section devoted to explaining the hukou system.
Response: Thanks for this comment. We have added a paragraph to discuss the impact of the hukou system (please see page 4, line 156-167)
6 The declaration in line 200 about the difficulty in the social integration of migrants is not rooted in the literature. Please add some empirical discussion that supports your view.
Response: Thanks for your comment. We explained the difficulty in the social integration of migrants from four aspects: institutional exclusion, labor market exclusion, exclusion in social relationships, and dis-crimination and stigmatization. Then, we separately discussed the four aspects based on many previous studies in the following paragraphs. Please refer to line 205-259. (please see page 5-6)
7 Line 268 – Did you mean to write ‘socially’ excluded? Make the corrections.
Response: We greatly appreciate this comment. Our original expression is indeed inappropriate. We have changed ‘social’ to ‘socially’. (please see page 6, line 276)
8 Line 281 – Rephrase to maximize their income.
Response: We appreciate this insightful comment. We have rephrased the sentence to ‘maximize their income’. (please see page 6, line 290)
9 After the observations you made from line 303, what does your study hope to add to correct the imbalance in studies with regard to your observations? This should come next before you start making further revelations of shortcomings in the study of urban-rural migration.
Response: We greatly appreciate this comment. We have specifically corrected this issue in the revised manuscript. The imbalance in attention allocation is not conducive to a systematic understanding of the social integration of rural-urban migrants. Therefore, this article attempts to establish an integrated conceptual framework to include different dimensions such as socioeconomic integration, acculturation, identity, civic engagement, political integration, and community participation. (please see page 7, line 313-317)
10 The sentence from 304-to 308 should be broken up into two or more sentences to improve their readability.
Response: Thanks for pointing this out. We have broken the sentence from 304 to 308 into two sentences. The modification improves their readability. (please see page 7, line 318-323)
11 Line 308 about the study by CGSS in 2012-2013 is not recent; rather authors may say that ‘the most recent CGSS study in 2012-2013.’ Recast that line.
Response: We appreciate this insightful comment. According to this comment, we have changed ‘the recent CGSS study in 2012–2013’ to ‘the most recent CGSS study in 2012–2013’. (please see page 7, line 317)
12 Please also consider to cite the following article in the Discussion and Conclusion: Khalid, B., & UrbaÅ„ski, M. (2021). Approaches to understanding migration: A multi-country analysis of the push and pull migration trend. Economics & Sociology, 14(4), 242–267. https://doi.org/10.14254/2071-789x.2021/14-4/14; Máté, D., Sarihasan, I., & Dajnoki, K. (2017). The relations between labour market institutions and employment of migrants. Amfiteatru Economic, 19(46), 806–820.
Response: Thank you for your recommendation. According to your suggestion, we have properly cited these articles in this revised manuscript. (please see page 13-14, line 597-599 & 649-652)
13 The entire study requires extensive editing and proofreading to improve the grammar and flow of the manuscript.
Response: Thank you for your significant recommendation. We have corrected the above grammatical errors and made an effort to correct the spelling and grammar errors and polish the whole manuscript. We also had the manuscript professionally proof read and edited to ensure it is accurate and understandable to readers. We have attached the certificate of English language editing in our revised manuscript submission.
Round 2
Reviewer 2 Report
The authors have elaborated some sections to make it more clear in the revised manuscript as per the suggestions; however, it would be nice if the authors focus a bit more on how to create readers' motivation to read their paper.
Author Response
We gratefully appreciate for your valuable comment. In response to your comment, we have made the following improvements:
First, we add in the abstract that this study contributes to improving the understanding of how to facilitate social integration of internal migrants in developing countries. (Please see page 1, line 23-24)
Second, we also emphasize in the Introduction section that the proposed framework of this study could be generalizable to the urban integration of internal migrants in other social settings, because the categorization of the process of rural–urban migration and understanding the role of governments are common issues during rural–urban integration. (Please see page 2, line 60-64)